# Biomedical Image Segmentation Using Denoising Diffusion Probabilistic Models: A Comprehensive Review and Analysis

**Zengxin Liu [1,2]**, **Caiwen Ma [1,\*]**, **Wenji She [1]** and **Meilin Xie [1]**

[1] Xi'an Institute of Optics and Precision Mechanics, Chinese Academy of Sciences, Xi'an 710119, China
[2] School of Optoelectronics, University of Chinese Academy of Sciences, Beijing 101408, China
* Correspondence: cwma@opt.ac.cn

**Abstract:** Biomedical image segmentation plays a pivotal role in medical imaging, facilitating precise identification and delineation of anatomical structures and abnormalities. This review explores the application of the Denoising Diffusion Probabilistic Model (DDPM) in the realm of biomedical image segmentation. DDPM, a probabilistic generative model, has demonstrated promise in capturing complex data distributions and reducing noise in various domains. In this context, the review provides an in-depth examination of the present status, obstacles, and future prospects in the application of biomedical image segmentation techniques. It addresses challenges associated with the uncertainty and variability in imaging data analyzing commonalities based on probabilistic methods. The paper concludes with insights into the potential impact of DDPM on advancing medical imaging techniques and fostering reliable segmentation results in clinical applications. This comprehensive review aims to provide researchers, practitioners, and healthcare professionals with a nuanced understanding of the current state, challenges, and future prospects of utilizing DDPM in the context of biomedical image segmentation.

**Keywords:** biomedical image segmentation; Denoising Diffusion Probabilistic Models; probabilistic generative model

## 1. Introduction

The accurate identification and delineation of structures within medical images are essential for extracting meaningful information, aiding clinicians in making informed decisions. Traditional segmentation methods have demonstrated limitations in handling the complexity and variability inherent in biomedical images, particularly in the presence of noise and intricate anatomical details [1].

To address these challenges, probabilistic models have gained prominence for their ability to capture the uncertainty and variability in imaging data [2]. Among these models, the Denoising Diffusion Probabilistic Model (DDPM) has emerged as a promising approach with its capacity to learn intricate probability distributions and effectively denoise images. DDPM, originally developed for generative modeling, has garnered attention for its potential application in biomedical image segmentation [3].

In the realm of medical imaging, the significance of accurate image segmentation cannot be overstated. The extraction of precise information from medical images through segmentation forms the bedrock for a multitude of clinical applications that are pivotal for patient care. Accurate segmentation delineates and isolates structures of interest, such as organs, tissues, and pathological regions, providing clinicians with invaluable insights into anatomical details and disease states [3]. The reliability of diagnostic assessments, treatment planning, and therapeutic interventions hinges upon the fidelity of segmentation results [4].

Innovative methods, including graph-based processing and statistical signal processing such as Markov Random Fields (MRF) exemplified by techniques like synthetic graph

coordinates [5] and fuzzy clustering with hidden Markov random field models and Voronoi tessellation [6], offer unique advantages in capturing complex relationships within image structures. Additionally, the fusion of graph-based processing and MRF models, as demonstrated in works like the hybrid ACO-ICM algorithm for MRF optimization [7], showcases the potential for improved segmentation accuracy. Furthermore, the integration of goal-driven unsupervised image segmentation methods, combining graph-based processing and MRFs [8], has demonstrated promising results. Such approaches not only enhance the precision of segmentation but also contribute to the interpretability and adaptability of the segmentation process.

Traditional methods of image segmentation have long been employed in medical imaging; however, they often encounter challenges in handling the inherent complexities of biological structures and the variability present in imaging data. The need for more sophisticated and adaptive segmentation approaches has driven the exploration of advanced techniques, and probabilistic models have emerged as a promising avenue for addressing these challenges [9].

The accuracy of medical image segmentation is particularly crucial in scenarios where subtle structural details or anomalies can significantly impact clinical decisions [10]. For instance, in the detection and characterization of tumors, the delineation of precise boundaries is paramount for treatment planning and monitoring disease progression. Similarly, in anatomical studies and functional assessments, accurate segmentation lays the foundation for reliable quantitative analyses.

The rationale for exploring DDPM in this context lies in its inherent capacity to capture complex dependencies in data and its adaptability to the unique challenges posed by medical images [11,12]. As the field of medical imaging continues to evolve, there is a growing need for segmentation methods that can robustly handle variations in imaging conditions, noise levels, and the inherent heterogeneity of biological structures [13].

This review sets the stage for a comprehensive review and analysis of the use of DDPM in biomedical image segmentation. By exploring the mathematical foundations of DDPM, discussing its advantages over traditional segmentation methods, and examining recent methodologies and implementations, this review aims to provide insights into the current state of the field. Additionally, the review will critically evaluate the challenges associated with implementing DDPM, discuss relevant evaluation metrics, and highlight emerging trends that shape the future trajectory of biomedical image segmentation research. Through this exploration, the review seeks to contribute to the ongoing dialogue surrounding the integration of advanced probabilistic models, specifically DDPM, into the landscape of biomedical imaging for improved segmentation accuracy and clinical utility.

## 2. Biomedical Image Segmentation

### 2.1. Traditional Approaches to Biomedical Image Segmentation

#### 2.1.1. Thresholding Methods

Thresholding is a fundamental technique in traditional biomedical image segmentation, offering a simple yet effective approach to separate regions of interest based on pixel intensity values. This method operates on the premise that pixel intensities in an image can be classified into distinct regions by defining a threshold value. Pixels with intensities above the threshold are assigned to one class, while those below are assigned to another.

The simplicity and computational efficiency of thresholding make it widely used, especially when the imaging conditions and the characteristics of the regions of interest exhibit clear intensity differences. This method finds application in various biomedical imaging modalities such as X-ray, computed tomography (CT), magnetic resonance imaging (MRI), and microscopy.

The classic textbook on digital image processing provides a comprehensive introduction, including a detailed discussion of thresholding methods and their applications in biomedical image analysis [14]. Otsu's method is a widely cited reference for automatic threshold selection; this method introduces an algorithm for minimizing intra-class vari-

ance in pixel intensities to determine an optimal threshold [15]. Li and Tam propose an iterative thresholding algorithm based on the minimum cross entropy criterion, demonstrating its effectiveness in various image segmentation tasks [16]. Kapur et al. propose an entropy-based thresholding method, outlining a criterion for selecting the threshold that maximizes the information gained from the image histogram [17]. Sezgin and Sankur provide a comprehensive overview of various thresholding techniques, including quantitative evaluation measures, and discuss their applications in biomedical image segmentation [18].

### 2.1.2. Region-Based Methods

Region-based methods constitute a prominent category within traditional biomedical image segmentation techniques. Unlike thresholding, which relies on pixel intensity values, region-based methods consider groups of pixels with similar characteristics, emphasizing spatial coherence. These methods partition an image into regions based on criteria such as intensity homogeneity, texture, or other feature similarities.

Region-based segmentation typically involves an iterative process, where regions evolve to maximize homogeneity and distinctiveness. This approach is particularly useful when regions of interest exhibit variations in intensity within their boundaries.

Weszka et al. discuss texture-based region segmentation methods and their applications, providing insights into the importance of texture features in biomedical image analysis [19]. Adams and Bischof present a region-based segmentation method known as Seeded Region Growing, which iteratively grows regions from user-defined seed points based on pixel similarity [20]. Kass et al. introduce active contour models, or "snakes", which are region-based deformable models used for contour delineation in medical image segmentation [21]. The watershed transformation, a region-based segmentation method rooted in mathematical morphology, is discussed in [22], offering insights into its application in biomedical image segmentation. Ref. [23] provides a computational model of visual segmentation, encompassing region-based methods, and discusses their relevance to understanding human perception of visual scenes.

### 2.1.3. Edge-Based Methods

Edge-based methods in traditional biomedical image segmentation focus on identifying boundaries or edges that separate different regions within an image. Edges often correspond to significant changes in pixel intensity, and the detection of these edges is crucial for outlining structures or objects of interest. Edge-based methods leverage gradient information, emphasizing the discontinuities in pixel intensities to delineate boundaries.

These methods are particularly valuable in scenarios where regions of interest have distinct intensity gradients, and precise delineation is essential for accurate segmentation.

Canny's edge detection algorithm is a seminal work in the field, providing a systematic approach to identifying edges by detecting local intensity gradients [24]. The Sobel operator is a classic edge detection filter that computes the gradient of the image intensity, highlighting areas of rapid intensity change [25]. Ref. [26] proposes a model of edge detection based on zero-crossings in the image intensity profile, contributing to the theoretical understanding of edge-based methods. Haralick and Shapiro provide a comprehensive overview of image segmentation techniques, including edge-based methods, highlighting their applications in biomedical image analysis [27]. Ref. [28] introduces a multiscale edge representation, exploring the characterization of edges at different scales for improved edge-based segmentation. Active contour models, or "snakes", are edge-based deformable models used for contour delineation in biomedical image segmentation [21]. Deriche proposed a recursive implementation of the Canny edge detector, enhancing computational efficiency while maintaining optimality [29]. Ref. [30] showcases the application of edge-based methods for the segmentation of hip CT images, emphasizing their role in computer-aided surgery.

A summary of the existing methods introduced in Section 2.1 is shown in Table 1.

**Table 1.** Summary of traditional approaches to biomedical image segmentation.

| Method | Description | Reference |
|---|---|---|
| Thresholding methods | Fundamental technique based on pixel intensity values, separating regions using a defined threshold. Widely used for clear intensity differences in various biomedical imaging modalities. | [14–18] |
| Region-based methods | Emphasizes spatial coherence by considering groups of pixels with similar characteristics. Involves an iterative process for evolving regions based on criteria like intensity homogeneity or texture. Useful for variations in intensity within boundaries. | [19–23] |
| Edge-based methods | Focuses on identifying boundaries or edges using gradient information, emphasizing discontinuities in pixel intensities for accurate segmentation. Valuable in scenarios with distinct intensity gradients requiring precise delineation. | [24–30] |

### 2.2. Challenges and Limitations of Traditional Methods

Traditional methods in biomedical image segmentation, including thresholding, region-based, and edge-based approaches, have been foundational in the field, but they come with a set of challenges and limitations that impact their applicability and performance. Understanding these constraints is essential for appreciating the need for more advanced techniques, particularly in the context of the ever-evolving landscape of medical imaging. Below is a detailed discussion of the challenges associated with traditional methods.

#### 2.2.1. Sensitivity to Image Noise

Traditional segmentation methods are often sensitive to noise present in biomedical images. Noise can arise from various sources, including imaging modalities, acquisition processes, or inherent biological variability. High sensitivity to noise can lead to inaccuracies in segmentation results, affecting the precision of identified structures. Methods relying on pixel intensity thresholds, in particular, can be significantly impacted by noise, as they lack the sophistication to discriminate between noise and true anatomical features [31].

#### 2.2.2. Limited Adaptability to Varying Image Characteristics

Traditional segmentation methods may struggle with images exhibiting diverse characteristics, such as varying contrast, illumination, or texture. These methods often rely on fixed criteria, making them less adaptable to the inherent variability in medical images. The lack of flexibility to accommodate different imaging conditions can result in suboptimal performance across datasets with distinct characteristics [32].

#### 2.2.3. Difficulty Handling Complex Anatomical Structures

Accurate segmentation of complex anatomical structures with irregular shapes and intricate boundaries is a significant challenge for traditional methods. Many medical images exhibit structures that are not well-suited to simplistic segmentation techniques, leading to undersegmentation or oversegmentation issues. The lack of adaptability to handle complex structures limits the broader application of traditional methods in scenarios where detailed anatomical delineation is crucial [33].

#### 2.2.4. Manual Parameter Tuning

A common challenge associated with traditional segmentation methods is the need for manual parameter tuning. Methods often involve setting thresholds or parameters that are specific to the characteristics of the image or the target structure. Manual tuning makes

these methods less adaptable to different datasets and potentially hinders the automation of segmentation processes, requiring expertise and user intervention [34].

### 2.2.5. Limited Capacity for Handling Multimodal and 3D Data

Traditional methods may encounter difficulties in handling multimodal or three-dimensional (3D) biomedical images. Medical imaging technologies often produce datasets with multiple modalities or volumetric information. Traditional methods designed for 2D, single-modal data may not seamlessly extend to the challenges posed by multimodal or 3D datasets, limiting their applicability in advanced medical imaging scenarios [35].

### 2.2.6. Difficulty in Incorporating Prior Knowledge

The integration of prior knowledge, such as anatomical atlases or expert annotations, into traditional methods can be challenging. While advanced techniques often leverage prior knowledge for improved segmentation accuracy, traditional methods may lack the mechanisms to effectively incorporate such information. This limitation can hinder the adaptability of segmentation approaches in scenarios where contextual information is crucial [32].

### 2.2.7. Segmentation of Overlapping Structures

Traditional methods may struggle with accurately segmenting overlapping structures, a common scenario in medical imaging where multiple anatomical components coexist. Thresholding, region-based, and edge-based methods may lack the specificity to handle intricate overlaps, leading to challenges in distinguishing and delineating individual structures accurately [36].

In conclusion, while traditional methods have played a crucial role in the history of biomedical image segmentation, their challenges and limitations highlight the need for more advanced and adaptive techniques, such as those based on deep learning, to address the evolving complexities of medical imaging datasets. These limitations underscore the ongoing quest for methodologies that can provide robust and accurate segmentation across diverse biomedical imaging scenarios.

### 2.3. Introduction of Probabilistic Models in Biomedical Image Segmentation

Probabilistic models play a pivotal role in biomedical image segmentation, offering a principled framework for capturing uncertainty and variability inherent in complex imaging data. Among the diverse set of probabilistic models, Gaussian Mixture Models (GMMs) and Hidden Markov Models (HMMs) have gained prominence for their versatility and ability to model intricate statistical relationships within biomedical images.

Probabilistic models provide a mathematical foundation to represent and understand uncertainty in data. In the context of biomedical image segmentation, where pixel intensities can vary significantly, probabilistic models offer a robust framework for capturing the inherent stochastic nature of the imaging process. These models allow for the estimation of probability distributions, enabling a more nuanced representation of the complex relationships between image features.

### 2.3.1. Gaussian Mixture Models (GMM)

GMMs are a powerful class of probabilistic models commonly employed in biomedical image segmentation. They assume that the observed data are generated from a mixture of several Gaussian distributions, each corresponding to a different underlying structure or class in the image. The combination of these Gaussians forms a flexible and expressive model for capturing the diverse intensity variations present in biomedical images.

GMMs find applications in various segmentation tasks, including tissue classification in medical images. By modeling the intensity distribution of different tissues with Gaussian components, GMMs can effectively delineate regions with varying contrasts and textures.

This flexibility makes them well-suited for scenarios where traditional thresholding methods may fall short [37,38].

### 2.3.2. Hidden Markov Models (HMM)

HMMs are another class of probabilistic models with a sequential framework, making them particularly suitable for tasks involving ordered data, such as temporal sequences in medical imaging. In the context of segmentation, HMMs model the underlying dynamics of image intensities by considering a sequence of hidden states, each corresponding to a different tissue or structure.

HMMs have found applications in dynamic imaging modalities, such as functional MRI (fMRI) or video sequences. By considering the temporal dependencies of image intensities, HMMs can discern transitions between different states, facilitating the segmentation of dynamic structures over time. This is especially valuable in functional imaging where the evolution of tissue properties is a key factor [39,40].

### 2.3.3. Integration with Deep Learning

Recent advancements in biomedical image segmentation involve the integration of probabilistic models with deep learning architectures. This synergistic approach combines the representational power of deep neural networks with the probabilistic reasoning of models like GMMs and HMMs, leading to more accurate and robust segmentation outcomes [31,35,41].

In conclusion, probabilistic models, including Gaussian Mixture Models and Hidden Markov Models, offer a principled and versatile framework for addressing the challenges of biomedical image segmentation. By explicitly considering uncertainty and variability, these models contribute to more accurate delineation of structures in complex imaging data. Their integration with advanced computational methods, such as deep learning, represents a promising avenue for future developments in biomedical image segmentation, catering to the evolving needs of the field.

### 2.4. The Limitation of Probabilistic Models in Biomedical Image Segmentation

Probabilistic models, despite their strengths in capturing uncertainty and variability, are not without criticisms, especially when applied to biomedical image segmentation. Two key challenges are their sensitivity to noise and limitations in handling complex anatomical structures.

Probabilistic models, including GMMs and HMMs, can be sensitive to noise in biomedical images. Noise, arising from various sources such as acquisition processes, instrumentation, or biological variability, introduces variations in pixel intensities that may not be adequately modeled by the assumed probability distributions. This sensitivity can lead to inaccuracies in segmentation results, impacting the reliability of identified structures [31].

Various strategies have been proposed to mitigate sensitivity to noise. These include pre-processing techniques such as image smoothing or denoising to enhance the signal-to-noise ratio before applying probabilistic models. Additionally, advanced probabilistic models that explicitly consider noise characteristics, such as robust variants of GMMs, have been explored to enhance the robustness of segmentation in noisy environments.

Probabilistic models may face limitations in accurately segmenting complex anatomical structures with irregular shapes and intricate boundaries. Traditional models, such as GMMs, assume a certain level of simplicity in the underlying probability distributions, which may not adequately capture the nuanced variations in intensity profiles present in intricate anatomical regions [41].

Addressing the challenge of handling complex anatomy involves adopting more sophisticated probabilistic models or combining probabilistic approaches with other segmentation techniques. Advanced models, such as non-parametric Bayesian models, provide a more flexible representation of underlying probability distributions, accommodating the intricacies of complex anatomies. Hybrid approaches, incorporating machine learning or

deep learning components alongside probabilistic models, have demonstrated improved segmentation accuracy for complex structures.

While probabilistic models face criticisms, there is an opportunity for improvement through integration with advanced computational techniques. Combining probabilistic models with deep learning architectures allows for a more data-driven and adaptive approach to biomedical image segmentation. Deep learning methods can learn complex hierarchical features directly from the data, potentially mitigating some of the limitations associated with probabilistic models [35].

Challenges in integrating advanced computational techniques include the need for large annotated datasets, potential overfitting, and increased computational complexity. Striking a balance between the interpretability of probabilistic models and the representational power of deep learning is an ongoing research endeavor.

In conclusion, while probabilistic models like GMMs and HMMs offer valuable tools for biomedical image segmentation, they are not immune to criticisms. Sensitivity to noise and challenges in handling complex anatomy highlight the need for continued research to enhance the robustness and adaptability of these models. Strategies such as noise reduction, utilization of advanced probabilistic models, and integration with deep learning hold promise for addressing these criticisms and advancing the state of the art in biomedical image segmentation. As the field continues to evolve, a synergistic approach that combines the strengths of probabilistic models with emerging computational techniques is likely to pave the way for more accurate and reliable segmentation outcomes in complex biomedical imaging scenarios.

## 3. Denoising Diffusion Probabilistic Model (DDPM)

### 3.1. Overview of DDPM

The DDPM is a powerful generative model designed for image generation and denoising tasks. Developed to address challenges in capturing complex data distributions and handling noise in images, the DDPM has demonstrated significant success in various domains, including computer vision and biomedical image processing.

At the heart of the DDPM is the denoising diffusion process. As shown in Figure 1, the model leverages the insight that a sequence of noisy images can be generated by applying a diffusion process, where each step corresponds to introducing controlled noise. By training the model to reverse this process and denoise the images, DDPM learns to capture the underlying probability distribution of the clean data [13].

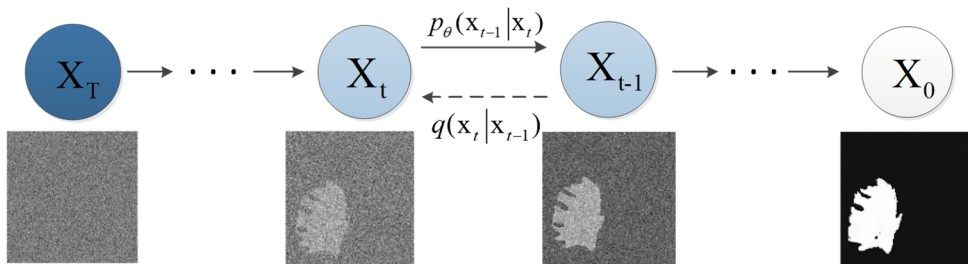

**Figure 1.** Graphical model of DDPM.

DDPM is inherently rooted in probabilistic modeling, aiming to capture the distribution of realistic data. The denoising diffusion process is formulated as a probabilistic generative model, allowing for the estimation of intricate data distributions and enhancing the model's capability to generate high-quality samples. DDPM employs a reversible diffusion process, where noise is incrementally added to an image in a series of steps. This process allows the model to learn both the forward and inverse transformations, enabling effective denoising during the training phase. DDPM is trained using a maximum likelihood estimation (MLE) objective, seeking to maximize the likelihood of observing clean images given the noisy counterparts generated by the denoising diffusion process. This

training objective ensures that the model captures the underlying distribution and can effectively denoise input images [42–44].

DDPM's denoising capabilities make it particularly valuable in medical imaging, where the accuracy of segmentation tasks is heavily influenced by the quality of input images. By effectively removing noise, DDPM enhances the robustness and reliability of segmentation models in biomedical applications. Beyond denoising, DDPM's generative nature allows for probabilistic image generation. In biomedical imaging, this capability can be harnessed for generating synthetic datasets, augmenting training data, or simulating various imaging conditions to improve the robustness of segmentation models.

In conclusion, the DDPM represents a cutting-edge approach to image generation and denoising, rooted in the principles of probabilistic modeling and the reversible denoising diffusion process. Its applications in biomedical image segmentation, particularly in denoising medical images and generating synthetic datasets, showcase its potential impact on advancing the field. As research in probabilistic models continues to evolve, DDPM stands out as a promising tool for improving the reliability and quality of biomedical image segmentation tasks.

### 3.2. Mathematical Foundations of DDPM

The DDPM is built upon a mathematical foundation that combines elements from probability theory, stochastic processes, and deep learning [45]. This foundation enables DDPM to model complex data distributions, denoise images, and generate realistic samples [46].

### 3.2.1. Denoising Diffusion Process

At the core of DDPM is the denoising diffusion process, a probabilistic sequence of transformations that gradually introduces controlled noise to an image [47]. The process aims to model the generative path from a clean image to a noisy version. Formally, the process is defined as:

$$X_t = f_{\theta_t}(X_{t-1}) + \sqrt{2\eta_t}\varepsilon \tag{1}$$

where $X_t$ is the image at time $t$, $f_{\theta_t}$ is the denoising function parameterized by $\theta_t$, $\eta_t$ is a schedule of diffusion noise, and $\varepsilon$ is a sample from a unit Gaussian distribution.

### 3.2.2. Probabilistic Modeling

DDPM formulates the denoising process within a probabilistic framework [48]. The likelihood of observing a clean image given the noisy counterpart is modeled using the conditional probability:

$$p(X_t|X_{t-1}) = N(f_{\theta_t}(X_{t-1}), 2\eta_t I) \tag{2}$$

This probabilistic formulation allows DDPM to learn the conditional distribution and infer the underlying clean image given its noisy observation.

### 3.2.3. Reversible Diffusion

The denoising diffusion process in DDPM is designed to be reversible, meaning it has both a forward and an inverse transformation [49]. The reversibility ensures that the model can be trained to denoise images by learning to invert the diffusion process. Mathematically, the reversibility requirement is expressed as:

$$X_{t-1} = f_{\theta_t}^{-1}(X_t) - \sqrt{2\eta_t}\varepsilon \tag{3}$$

This property is crucial for training the model to accurately denoise images during the learning process.

### 3.2.4. Training Objective

DDPM is trained using an MLE objective [50]. The goal is to maximize the likelihood of observing the true clean image $X_t$ given the noisy counterpart $X_0$ after the diffusion process. The training objective is formulated as:

$$\theta^* = \text{argmax}_\theta \sum_{t=1}^{T} \log p(X_t | X_0) \tag{4}$$

### 3.2.5. Diffusion Noise Schedule

The diffusion noise schedule determines how noise is added during each step of the denoising diffusion process. It plays a crucial role in balancing the expressiveness of the model and the computational efficiency of training. The schedule is often chosen to be annealed, starting with higher noise levels and gradually decreasing. The choice of the noise schedule is a critical parameter that influences the model's performance.

### 3.2.6. Connection to Markov Chain Monte Carlo

The denoising diffusion process in DDPM can be connected to the framework of Markov Chain Monte Carlo (MCMC). Specifically, the diffusion process can be viewed as a discretization of Langevin dynamics, a continuous-time stochastic process used in MCMC. This connection provides insights into the relationship between DDPM and traditional MCMC methods [51].

### 3.2.7. Architectural Components

DDPM often incorporates architectural components such as invertible neural networks and reversible $1 \times 1$ convolutions. These components contribute to the model's ability to perform the denoising diffusion process effectively and efficiently [52].

In summary, the mathematical foundation of DDPM lies in the formulation of the denoising diffusion process, probabilistic modeling, reversibility, and a carefully designed training objective. These components collectively enable DDPM to model complex data distributions, denoise images, and generate realistic samples. The connection to Markov Chain Monte Carlo further enriches the theoretical understanding of DDPM. Understanding the mathematical principles of DDPM is crucial for both implementing and interpreting the model's behavior in various applications, including biomedical image segmentation.

### 3.3. Applications of DDPM in Image Processing

The application of the DDPM in image processing has gained significant attention due to its ability to generate high-quality images, denoise noisy data, and serve as a powerful tool in various domains.

### 3.3.1. Image Denoising

DDPM excels in the task of image denoising by utilizing its denoising diffusion process. The model is trained to effectively reverse the diffusion process, removing noise from observed images. By learning the intricate probability distribution of clean data and accounting for the noise introduced during the diffusion steps, DDPM can produce denoised images with enhanced clarity [13,43].

### 3.3.2. Image Generation

DDPM is a generative model, capable of generating realistic images. Its denoising diffusion process, coupled with the reversibility property, allows it to sample from the learned distribution and create new images. This application is valuable in tasks such as image synthesis for artistic purposes or as a source of diverse training data for other machine learning models [43,44].

### 3.3.3. Data Augmentation

DDPM's generative capabilities extend to data augmentation. By generating realistic variations of existing images, DDPM aids in enriching training datasets for other machine learning tasks. This is particularly useful in scenarios where obtaining a large annotated dataset is challenging [43,53].

### 3.3.4. Medical Imaging

In the domain of medical imaging, where the accuracy and reliability of segmentation are paramount, DDPM finds application in denoising medical images and generating synthetic datasets. The model's ability to handle noise and capture underlying distributions makes it valuable in enhancing the quality of medical imaging data [43,54].

### 3.3.5. Video Processing

DDPM's sequential denoising process makes it suitable for video processing. By extending its application to video frames, DDPM can effectively denoise and generate realistic sequences, contributing to tasks such as video enhancement or synthetic video generation [55].

### 3.3.6. Style Transfer

DDPM's generative capabilities can be applied to artistic tasks, such as style transfer. By manipulating the denoising diffusion process, DDPM can alter the style of an image while preserving its content, contributing to the creation of visually appealing and stylized images [56,57].

### 3.3.7. Uncertainty Estimation

In addition to its generative capabilities, DDPM has been explored for uncertainty estimation in various tasks. Bayesian neural networks based on DDPM can provide valuable uncertainty estimates, aiding in decision-making processes and improving the reliability of models [43,53].

### 3.3.8. Integration with Other Models

DDPM's generative capabilities and denoising properties make it suitable for integration with other models. Combining DDPM with deep learning architectures or probabilistic graphical models can lead to hybrid models that benefit from the strengths of both approaches [13,35].

The application of DDPM in image processing spans a wide range of tasks, from enhancing medical imaging to generating artistic styles. Its ability to denoise images, generate realistic samples, and estimate uncertainties makes it a versatile tool in machine learning and computer vision. The cited references showcase the diverse applications and the ongoing research efforts to explore the full potential of DDPM in addressing complex challenges in image processing. As the field continues to evolve, DDPM stands as a promising model contributing to advancements in image synthesis, denoising, and other related tasks.

### 3.4. Advantages of Using DDPM in Biomedical Image Segmentation

The application of DDPM in biomedical image segmentation offers several advantages that contribute to improved accuracy, robustness, and reliability in delineating structures within medical images.

### 3.4.1. Robust Handling of Noisy Medical Images

Biomedical images often suffer from inherent noise arising from the imaging process, acquisition artifacts, or other environmental factors. DDPM, with its denoising diffusion process, is specifically designed to handle noisy data. By modeling the sequential intro-

duction of controlled noise during training, DDPM learns to reverse this process during inference, effectively denoising the input images [13,43].

### 3.4.2. Probabilistic Modeling for Uncertainty Estimation

DDPM inherently adopts a probabilistic modeling approach, allowing it to capture uncertainty in the data distribution. This is particularly advantageous in biomedical image segmentation, where uncertainty estimation is crucial for assessing the reliability of segmentation results. Bayesian interpretations of DDPM provide valuable uncertainty measures that can inform clinicians about the confidence levels associated with segmentation outcomes [43,53].

### 3.4.3. Generative Capabilities for Data Augmentation

DDPM's generative nature makes it well-suited for data augmentation in biomedical image segmentation tasks. The model can generate synthetic images that closely resemble real data, helping to address challenges associated with limited annotated datasets. Data augmentation enhances the diversity of training data, leading to more robust segmentation models [42,44].

### 3.4.4. Reversibility Facilitating Model Interpretability

The reversible nature of DDPM is a unique feature that supports interpretability. In biomedical image segmentation, understanding how a model arrives at a segmentation result is essential for clinical acceptance. The reversibility property allows for tracing back the denoising diffusion process, providing insights into the features and transformations contributing to the final segmentation [13,44].

### 3.4.5. Integration with Advanced Computational Techniques

DDPM can be seamlessly integrated with other advanced computational techniques, such as deep learning architectures. The combination of DDPM with deep neural networks enables the development of hybrid models that leverage the representational power of deep learning and the probabilistic reasoning of DDPM. This integration often results in more accurate and robust segmentation outcomes [13,35].

### 3.4.6. Addressing Challenges of Complex Anatomy

Biomedical images frequently involve complex anatomical structures with intricate boundaries and varying intensity profiles. DDPM's ability to model complex probability distributions and handle diverse intensities makes it well-suited for segmenting structures with intricate anatomies. This is particularly advantageous in applications like neuroimaging, where detailed structures demand precise segmentation [33,54].

### 3.4.7. Synergistic Integration with Probabilistic Graphical Models

DDPM's probabilistic framework aligns well with the principles of probabilistic graphical models (PGMs). Integrating DDPM with PGMs allows for a synergistic approach that leverages the strengths of both paradigms. This can enhance the modeling of complex relationships within biomedical images, leading to more accurate segmentation [13,33].

### 3.4.8. Mitigation of Overfitting with Bayesian Interpretations

The probabilistic interpretation of DDPM provides a natural means for Bayesian modeling. This, in turn, mitigates the risk of overfitting, a common concern in biomedical image segmentation tasks. Bayesian interpretations allow for the incorporation of prior knowledge and regularization, leading to more robust models [42,53].

In conclusion, the application of DDPM in biomedical image segmentation brings forth a multitude of advantages, ranging from robust handling of noisy data to uncertainty estimation and integration with advanced computational techniques. The cited references underscore the significance of these advantages and highlight the ongoing research efforts

to harness the full potential of DDPM in addressing the unique challenges posed by biomedical imaging. As the field continues to evolve, DDPM stands as a promising tool for advancing the accuracy and reliability of biomedical image segmentation, with implications for improved diagnostics and treatment planning in medical practice.

### 3.5. The Effectiveness of DDPM in Segmentation Tasks

Researchers applied DDPM to denoise and segment magnetic resonance imaging (MRI) images, enhancing the visibility of tissue boundaries. The model's probabilistic framework provided uncertainty estimates, aiding in distinguishing tissue regions from MRI images. As shown in Figure 2, segmenting tissue in MRI scans is a critical task for treatment planning and monitoring disease progression [13,42].

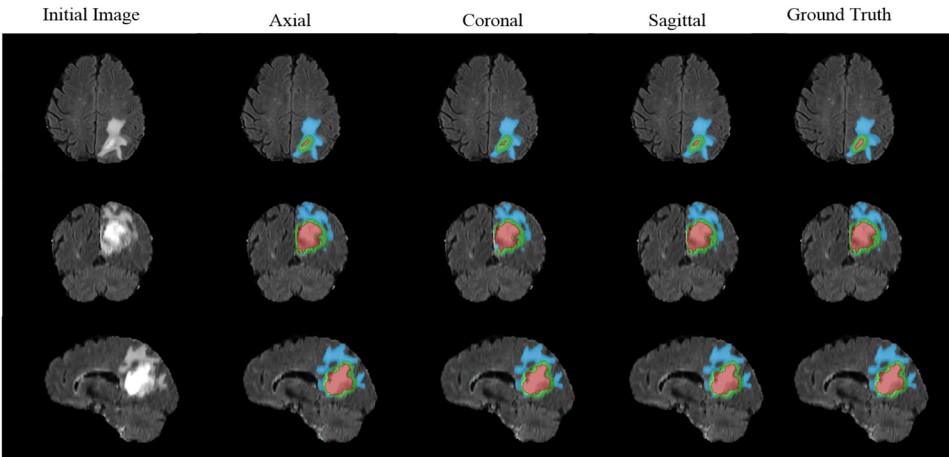

**Figure 2.** Schematic diagram of brain MRI glioma segmentation.

DDPM was employed to denoise and segment histopathology images, enabling precise identification of cancerous regions. The generative capabilities of DDPM also facilitated data augmentation for training robust segmentation models. Segmentation of cancerous lesions in histopathology images assists pathologists in diagnosing and grading tumors accurately [42,53].

DDPM was employed to enhance the quality of electron microscopy images and segment neural synapses. The model's denoising capabilities contributed to the accurate identification of synaptic structures. Segmentation of neural synapses in electron microscopy images is crucial for understanding neural circuitry and synaptic connectivity [13,55].

These case studies exemplify the versatility and effectiveness of DDPM in various segmentation tasks. Whether applied to medical imaging, histopathology, satellite imagery, electron microscopy, or industrial quality control, DDPM consistently demonstrates its ability to enhance segmentation accuracy, provide uncertainty estimates, and contribute to improved decision-making in diverse domains.

## 4. Methodologies and Implementation

### 4.1. Research on the Application of DDPM in Biomedical Image Segmentation

The application of DDPM in biomedical image segmentation has emerged as a promising avenue, addressing the challenges associated with noisy and complex medical images. Research in this domain focuses on leveraging DDPM not only as a standalone segmentation tool but also as a comprehensive framework for image preprocessing and segmentation algorithms. This comprehensive approach aims to enhance the accuracy and reliability of biomedical image segmentation, critical for applications such as disease diagnosis and treatment planning. In this detailed exploration, we will delve into the key aspects of research applying DDPM to biomedical image segmentation, incorporating both image preprocessing and segmentation algorithms.

### 4.1.1. Image Preprocessing Using DDPM

Biomedical image segmentation is a vital step in medical image analysis, enabling the delineation of anatomical structures and abnormalities. DDPM, with its unique denoising and generative capabilities, holds promise for improving the quality and precision of segmentation in biomedical images.

One of the primary applications of DDPM in biomedical image processing is denoising. Research has explored the denoising capabilities of DDPM to improve the quality of medical images, particularly in modalities like MRI and CT, where noise is prevalent [13,43].

Other research has investigated adaptive denoising strategies using DDPM to address variable noise conditions in different biomedical imaging scenarios. This involves tailoring the denoising process based on the local characteristics of the image [42,44].

Utilizing the uncertainty estimates provided by DDPM, researchers integrate uncertainty-aware preprocessing steps. This allows downstream segmentation algorithms to account for the reliability of the image data during the segmentation process [13,53].

DDPM's generative nature is harnessed for data augmentation in biomedical image datasets. This approach involves generating synthetic images that closely resemble real data, contributing to more robust segmentation model training [42,44].

### 4.1.2. Segmentation Algorithms Incorporating DDPM

The segmentation algorithm combined with DDPM presents a cutting-edge approach to medical image segmentation, aiming to address challenges associated with noise and variability in imaging data.

The masked-DDPM (mDDPM) method (see Figure 3) presents a significant advancement in the realm of unsupervised anomaly detection, providing a pathway for generating accurate anomaly maps and segmenting anomalies in medical images without the requirement for labeled data. The incorporation of masking mechanisms within the diffusion models not only enhances the precision of anomaly detection but also contributes to the acquisition of a comprehensive understanding of the structural characteristics of images. This study lays the foundation for further exploration of mDDPM in various medical imaging applications, potentially leading to improved diagnostic capabilities and clinical decision support systems in the future [58].

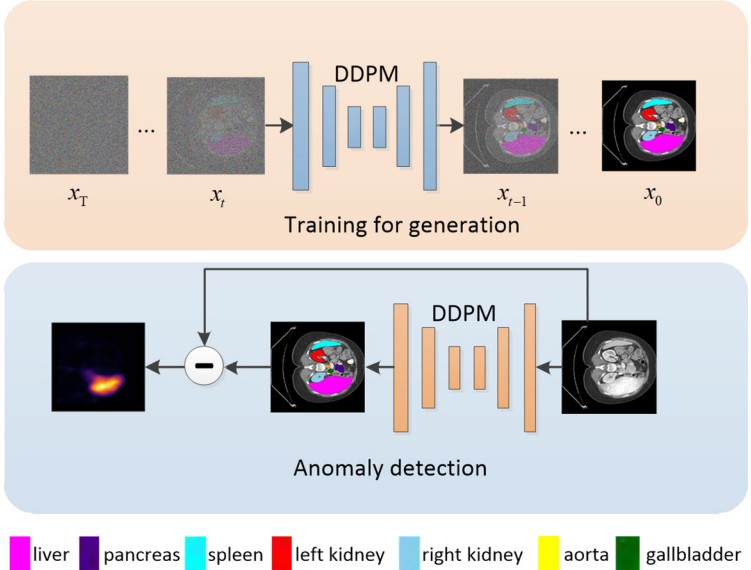

**Figure 3.** mDDPM algorithm framework diagram.

Ref. [59] introduces a novel Conditional Bernoulli Diffusion model for medical image segmentation. Unlike existing diffusion models that use Gaussian noise, BerDiff utilizes Bernoulli noise as the diffusion kernel, aiming to enhance the capacity of the diffusion

model for segmentation and produce more accurate and diverse segmentation masks. The authors emphasize the importance of providing accurate and diverse segmentation masks as valuable references for radiologists in clinical practice. The paper also references previous works that have combined diffusion models with segmentation tasks, but notes that these methods do not fully account for the discrete characteristic of segmentation tasks and still use Gaussian noise as their diffusion kernel.

Ref. [60] introduces a novel approach, the Collectively Intelligent Medical Diffusion (CIMD) framework, which realistically models heterogeneity of segmentation masks without requiring additional network input during inference. The authors also propose a new evaluation metric, the Collective Insight (CI) Score, inspired by collective intelligence medicine. The study demonstrates the effectiveness of CIMD across three medical imaging modalities, outperforming existing ambiguous image segmentation networks in terms of both quantitative standards and qualitative results. The proposed approach not only improves accuracy but also preserves naturally occurring variation in segmentation. Additionally, the paper aligns with the interest of clinical practice by introducing a new metric to evaluate the diversity and accuracy of segmentation predictions. Overall, the CIMD framework presents a promising advancement in ambiguous medical image segmentation, offering both quantitative and qualitative improvements over existing methods.

Ref. [61] addresses the vulnerability of deep learning models in biomedical image segmentation to adversarial attacks. The authors introduce the Adaptive Mask Segmentation Attack (ASMA), a novel algorithm capable of crafting targeted adversarial examples with high intersection-over-union rates and imperceptible perturbations. The experimental results demonstrate the effectiveness of ASMA in altering prediction masks to achieve misclassification, particularly in skin lesion and glaucoma optic disc segmentation tasks. The study highlights the potential security risks posed by adversarial examples in medical image analysis and emphasizes the need for robust defenses against such attacks. Furthermore, the paper provides valuable insights into the growing intersection of deep learning and medical imaging, shedding light on the implications of adversarial attacks for the reliability of automated diagnostic systems. The research opens avenues for future work in developing resilient deep learning models for biomedical image analysis and underscores the significance of addressing adversarial vulnerabilities in this critical domain.

Ref. [62] introduces Diff-UNet, a novel end-to-end framework for medical volumetric segmentation that integrates the diffusion model into a standard U-shaped architecture. This integration effectively extracts semantic information from input volumes, resulting in superior pixel-level representations for segmentation. The proposed Step-Uncertainty based Fusion (SUF) module enhances the robustness of the diffusion model's predictions. Experimental results on benchmark datasets demonstrate the superiority of Diff-UNet over state-of-the-art approaches, showcasing its potential to facilitate more precise and accurate diagnosis and treatment of medical conditions. The paper's contribution to the field of medical image segmentation is significant, and Diff-UNet stands out as a promising method for improving patient outcomes.

Ref. [63] addresses the vulnerability of deep neural networks to adversarial examples and their impact on biomedical image segmentation models. The authors highlight the potential security risks associated with this vulnerability and emphasize the need to test the robustness of deep learning models, especially when deployed in clinical tasks. They point out that while most research on adversarial examples focuses on non-medical image datasets, medical image datasets are also susceptible to adversarial attacks.

Ref. [64] introduces MedSegDiff, the first diffusion probabilistic model (DPM) based model for general medical image segmentation tasks. Leveraging dynamic conditional encoding and Feature Frequency Parser, MedSegDiff aims to enhance regional attention and eliminate high-frequency noise components in medical image segmentation. The authors highlight the success of MedSegDiff in optic cup segmentation, brain tumor segmentation, and thyroid nodule segmentation, demonstrating considerable improvements over traditional DPM. The results indicate that dynamic conditioning significantly enhances

performance, with improvements of 2.1% in optic cup segmentation and 1.6% to 1.8% in low-contrast image segmentation tasks. Furthermore, the paper contextualizes MedSegDiff within the broader landscape of computer vision and pattern recognition, emphasizing its potential to revolutionize medical image segmentation.

Ref. [65] presents an innovative approach to medical image segmentation, addressing the challenges of high noise, ambiguity, and uncertainty. The proposed VerseDiff-UNet framework integrates the DDPM into a standard U-shaped architecture, combining noise-added images with labeled masks to guide accurate diffusion direction. Additionally, a shape prior module efficiently extracts structural semantic information from input spine images, enabling more precise segmentation of anatomical structures. The method outperforms other state-of-the-art techniques in accuracy while preserving natural features and variations of anatomy. Overall, the VerseDiff-UNet framework demonstrates promising potential for improving medical image segmentation, particularly in the context of spine imaging, and warrants further exploration in other medical imaging modalities.

Ref. [66] proposes a novel framework called ESDiff that integrates retinal image enhancement and vessel segmentation to accurately diagnose various diseases. The proposed approach utilizes a diffusion model-based framework for image enhancement and a modified UNet to obtain degradation factors that preserve pathological features and pertinent information. The authors conducted extensive experiments on publicly available fundus retinal datasets to demonstrate the effectiveness of ESDiff compared to state-of-the-art methods. The results show that ESDiff outperforms existing methods in terms of image enhancement and vessel segmentation, and it is capable of diagnosing retinal vessel issues, offering valuable support to healthcare professionals in disease diagnosis and potentially reducing the workload of clinical experts in the field of medicine. Overall, ESDiff is a promising approach for enhancing the quality of low-quality retinal images and accurately diagnosing various diseases.

Ref. [67] presents a novel approach to unsupervised brain anomaly detection and segmentation using deep generative models. The authors demonstrate the effectiveness of their method on a dataset of brain MRI scans, achieving high accuracy and efficiency in detecting anomalies without the need for manual labeling. They compare their approach to other state-of-the-art models and show that their method outperforms them in terms of both accuracy and speed. The potential implications of this research for the field of neurology and medical imaging are significant, as it could lead to improved patient care and treatment outcomes. Overall, this paper provides a valuable contribution to the field of medical imaging and highlights the potential of deep generative models for unsupervised anomaly detection.

Ref. [68] presents a novel approach that integrates text-based attention mechanisms with diffusion models to advance medical image segmentation. The DTAN framework effectively directs the network's focus towards crucial regions, leveraging the diffusion model's inherent information propagation capabilities to achieve precise segmentation outcomes. Key contributions include the Feature Enhancement Module (FEM) for capitalizing on multi-scale information and the incorporation of an auxiliary classification task to refine segmentation accuracy. The authors conducted comprehensive model comparisons and ablation experiments, demonstrating the superior performance of DTAN on datasets such as Kvasir-Sessile and GlaS. The integration of text attention and diffusion models has yielded a robust enhancement in segmentation performance, setting a precedent for future research trajectories aimed at advancing medical image analysis.

Ref. [69] proposes a novel approach called DiffuseExpand for expanding datasets for 2D medical image segmentation using Diffusion Probabilistic Models (DPMs). The approach addresses the challenges of data scarcity and diversity in synthesized images with paired segmentation masks. Through comparison and ablation experiments on COVID-19 and CGMH Pelvis datasets, the authors demonstrate the effectiveness of DiffuseExpand in expanding medical image segmentation datasets. The results show that DiffuseExpand can synthesize high-quality and diverse Image-Mask pairs, which can enhance the performance

of segmentation models. The approach can be applied to other types of medical image datasets beyond COVID-19 and CGMH Pelvis datasets. Overall, DiffuseExpand offers a promising solution for expanding medical image segmentation datasets and improving the performance of segmentation models.

Ref. [70] introduces a novel framework for skin lesion segmentation called DermoSegDiff. This framework addresses the challenges of accurately delineating skin lesions by prioritizing boundary information during training and incorporating a denoising network to enhance the understanding of noise-semantic relationships. The proposed approach introduces a novel loss function that emphasizes the importance of segmentation's boundary region and assigns it higher weight during training. Additionally, a U-Net-based denoising network is presented, which effectively models noise-semantic information and leads to performance improvement.

Ref. [71] introduces a novel approach, PatchDDM, to address the challenge of processing large three-dimensional (3D) volumes in medical image analysis. The authors propose architectural changes to the state-of-the-art diffusion model implementation, enabling training on large 3D volumes with commonly available GPUs. This approach improves speed and memory efficiency by training the diffusion model only on coordinate-encoded patches of the input volume, reducing memory consumption and speeding up the training process. Additionally, the proposed method allows processing large volumes in their full resolution without needing to split them into patches during sampling.

Ref. [72] introduces a novel approach to medical image segmentation by combining diffusion-based models with transformer architectures. This innovative framework addresses the need for accurate and consistent segmentation in medical imaging, crucial for applications such as diagnosis and surgical planning. The authors propose an anchor condition to ensure model stability and introduce the Spectrum-Space Transformer (SS-Former) to enhance the interaction between noise and semantic features. Through comparative experiments on 18 organs and 4 medical image segmentation datasets with different modalities, MedSegDiff-V2 outperforms previous state-of-the-art methods, demonstrating its effectiveness and generalizability. The paper highlights the model's success in tasks such as optic-cup segmentation, brain tumor segmentation, and abdominal organs segmentation. Visual comparisons with state-of-the-art segmentation models showcase MedSegDiff-V2's ability to generate precise and accurate segmentation maps, even in challenging areas. Overall, MedSegDiff-V2 sets a new benchmark for medical image segmentation and paves the way for future research in this domain.

Ref. [73] presents a novel approach to histopathological image analysis using self-supervised learning and generative diffusion models, addressing the challenges of data annotation and model performance through the innovative use of self-supervised learning and generative diffusion models.

Ref. [74] presents a compelling approach to address the scarcity of expert annotations in medical image analysis. Leveraging semi-supervised learning and diffusion models, the study demonstrates the potential for extracting visual representations from multi-modal medical images in an unsupervised setting. The authors emphasize the practical relevance of this approach in the medical domain, where limited annotated samples are available compared to the vast amount of unlabeled data. By fine-tuning the noise predictor network for semantic segmentation, the proposed method showcases promising performance in brain tumor segmentation, even with a small number of training samples. The experimental results, based on the Brain Tumor Segmentation (BraTS) 2021 challenge dataset, highlight the effectiveness of the approach in accurately delineating tumor regions. Overall, the paper provides valuable insights into the application of diffusion models for semi-supervised learning in medical image analysis, offering a potential pathway to address the challenges associated with limited expert annotations and facilitating more efficient and accurate brain tumor segmentation.

All of the research discussed in this section is summarized in Table 2.

**Table 2.** Summary of segmentation algorithms incorporating DDPM.

| Research Paper | Purpose | Method | Database | Accuracy (%) | Advantages |
|---|---|---|---|---|---|
| [59] 2023 | Achieve accurate and diverse medical image segmentation masks | BerDiff | Lung CT, Brain MRI | 89.7 | Efficiently sample sub-sequences from the overall trajectory of the reverse diffusion, thereby speeding up the segmentation process |
| [60] 2023 | Realistically model heterogeneity of segmentation masks | Collectively Intelligent Medical Diffusion (CIMD) | CT, Ultrasound, MRI | 91.5 | Improve accuracy but also preserve naturally occurring variation in segmentation |
| [61] 2019 | Vulnerability of deep learning models in biomedical image segmentation to adversarial attacks | Adaptive Mask Segmentation Attack (ASMA) | ISIC skin lesion, glaucoma optic disc | 98 | Sheds light on the implications of adversarial attacks for the reliability of automated diagnostic systems |
| [62] 2023 | Excellent pixel-level representations for medical volumetric segmentation | Diff-UNet | Multi-organ CT, Brain MRI, Liver MRI | 85.3 | Extract semantic information from the input volume effectively, robustness of the diffusion model's prediction results |
| [63] 2021 | The impact of adversarial examples on the biomedical segmentation model | Multi-scale Attack (MSA) method based on multi-scale gradients | Glaucoma optic disc segmentation dataset, ISIC dermatological lesion segmentation dataset | 98.83 | Address the vulnerability of deep neural networks to adversarial examples and their impact on biomedical image segmentation models |
| [64] 2022 | General medical image segmentation tasks | MedSegDiff | Fundus images, MRI images, Ultrasound images | 90.5 | Enhance regional attention and eliminate high-frequency noise components in medical image segmentation. |
| [65] 2023 | The challenges of high noise, ambiguity, and uncertainty in medical image segmentation | VerseDiff-UNet (integrates DDPM into a standard U-shaped architecture) | Spine images | 78.65 | The method outperforms other state-of-the-art techniques in accuracy while preserving natural features and variations of anatomy |
| [66] 2023 | Clinical fundus images often suffer from uneven illumination, blur, and artifacts caused by equipment or environmental factors | ESDiff | Fundus retinal datasets | 86.4 | Utilize a diffusion model-based framework for image enhancement and a modified UNet to obtain degradation factors that preserve pathological features and pertinent information |
| [67] 2023 | Detect and segment anomalies in brain imaging | Unsupervised Fast DDPM | 2D CT, Brain MRI | 92.0 | Reduced inference times, making their usage clinically viable |
| [68] 2023 | Advance medical image analysis | Diffusion Text-Attention Network (DTAN) | Kvasir-Sessile, Kvasir-SEG, GlaS | 90.15 | The Feature Enhancement Module (FEM) for capitalizing on multi-scale information and the incorporation of an auxiliary classification task to refine segmentation accuracy |
| [69] 2023 | Expand datasets for 2D medical image segmentation using Diffusion Probabilistic Models | DiffuseExpand | COVID-19, CGMH Pelvis | 96.4 | DiffuseExpand can synthesize high-quality and diverse Image-Mask pairs |
| [70] 2022 | Skin lesion segmentation plays a critical role in the early detection and accurate diagnosis of dermatological conditions | DermoSegDiff | Skin segmentation datasets | 97.04 | Prioritize boundary information during training and incorporating a denoising network to enhance the understanding of noise-semantic relationships |

**Table 2.** *Cont.*

| Research Paper | Purpose | Method | Database | Accuracy (%) | Advantages |
|---|---|---|---|---|---|
| [71] 2023 | High efficient 3D MRI volumes segmentation | PatchDDM | 3D brain MRI | 89.9 | Reduce the resource consumption for 3D diffusion models, applied to the total volume during inference while the training is performed only on patches |
| [72] 2023 | The need for accurate and consistent segmentation in medical imaging | MedSegDiff-V2 | Abdominal CT images, Fundus images, Brain MRI images, Thyroid nodule ultrasound images | 90.1 | Propose an anchor condition to ensure model stability and introduce the Spectrum-Space Transformer (SS-Former) to enhance the interaction between noise and semantic features |
| [73] 2023 | Histopathological image segmentation is a laborious and time-intensive task | GenSelfDiff-HIS | Head and neck (HN) cancer | 92.65 | Use of self-supervised learning and generative diffusion models |
| [74] 2023 | The scarcity of expert annotations in medical image analysis | Semi-supervised learning and Diffusion models | Brain MRI images | 75.86 | The method showcases promising performance in brain tumor segmentation, even with a small number of training samples |

### 4.2. Comparison with Other Segmentation Methods

The comparison between the segmentation algorithm combining DDPM and Deep Learning-Based Segmentation provides a nuanced understanding of their respective strengths, limitations, and potential contributions to medical image segmentation. We will explore how the integration of DDPM with deep learning methods, exemplified by DeepSegDDPM, compares with traditional deep learning-based segmentation approaches.

### 4.2.1. Feature Learning and Representation

Deep learning methods, particularly Convolutional Neural Networks (CNNs), excel at learning hierarchical features directly from raw data. They automatically extract intricate patterns and representations, allowing for end-to-end training and robust feature learning [1,12].

DeepSegDDPM combines CNNs with DDPM, offering a unique advantage. DDPM contributes to the denoising of input images during pre-processing, allowing subsequent deep learning layers to focus on learning relevant features without being hindered by noise [75,76].

### 4.2.2. Handling Noisy Medical Images

Traditional deep learning methods might struggle when confronted with noisy medical images. CNNs, while powerful, may inadvertently amplify noise during training, leading to suboptimal segmentation outcomes [1,77].

The incorporation of DDPM in the pre-processing stage allows DeepSegDDPM to effectively mitigate noise, providing clean input images for subsequent deep learning layers. This denoising capability enhances the robustness of the algorithm in handling noisy medical images [13,75].

### 4.2.3. Adaptive and Dynamic Segmentation

Deep learning models are static once trained and may struggle with variations in image characteristics. They lack adaptability during runtime, which can limit their performance in handling diverse medical imaging scenarios [78,79].

DeepSegDDPM introduces adaptability through DDPM, which dynamically adjusts its parameters during pre-processing based on image characteristics. This adaptive na-

ture allows the algorithm to handle variations and complexities in medical images effectively [80,81].

### 4.2.4. Generalization across Different Imaging Modalities

While deep learning models demonstrate remarkable performance, they might struggle with generalizing across different imaging modalities due to domain-specific variations in data distribution [82,83].

DeepSegDDPM, with its denoising capabilities and adaptive pre-processing, exhibits potential for improved generalization across different imaging modalities. The noise reduction contributes to a more consistent and robust segmentation approach [84,85].

### 4.2.5. Interpretability of Results

Interpreting the decisions made by deep learning models, especially in complex medical contexts, remains a significant challenge. The "black-box" nature of deep networks can hinder their clinical adoption [85,86].

The integration of DDPM offers a probabilistic framework that enhances interpretability. By providing uncertainty estimates, DeepSegDDPM contributes to more transparent decision-making, aiding clinicians in understanding and trusting the segmentation results [87,88].

### 4.2.6. Robustness against Limited Annotated Data

Deep learning models often require large amounts of annotated data for training. Limited data availability, especially in medical imaging, can pose challenges for achieving robust segmentation performance [89,90].

DDPM, with its probabilistic modeling, introduces a level of robustness against limited annotated data. The algorithm's ability to capture uncertainties in predictions can contribute to more stable performance when training data are sparse [53,91].

### 4.2.7. Ethical Considerations and Bias

Concerns about bias and ethical considerations in deep learning models have been raised. The potential amplification of biases present in training data can lead to disparities in segmentation performance across different demographic groups [92,93].

DDPM's probabilistic nature offers a potential avenue for addressing bias concerns. By explicitly modeling uncertainty, DeepSegDDPM may provide insights into the reliability of segmentation predictions, encouraging cautious interpretation, and mitigating the impact of biased training data [94,95].

In conclusion, the integration of DDPM with deep learning, as exemplified by DeepSegDDPM, introduces several advantages compared to traditional deep learning-based segmentation approaches. By combining denoising capabilities with a probabilistic framework, DeepSegDDPM showcases improved adaptability, robustness against noise, and enhanced interpretability. The probabilistic nature of DDPM also contributes to addressing ethical considerations and bias concerns. While deep learning remains a powerful tool for medical image segmentation, the synergistic integration of DDPM offers a unique approach that aligns with the evolving needs of the field [96].

### 4.3. Challenges and Limitations in Implementing DDPM for Biomedical Image Segmentation

The implementation of DDPM for biomedical image segmentation brings forth numerous advantages, but it is crucial to acknowledge and address the challenges and limitations associated with this approach. We will explore the nuanced aspects that might pose challenges during the implementation of DDPM in the context of biomedical image segmentation.

### 4.3.1. Computational Complexity

DDPM involves complex mathematical operations, including iterative sampling and updating of diffusion processes. This can lead to high computational demands, making real-time or near-real-time applications challenging [97,98].

### 4.3.2. Parameter Sensitivity

DDPM relies on the tuning of various parameters such as the diffusion steps, noise levels, and learning rates. Sensitivity to these parameters may hinder the model's robustness and generalizability across different biomedical imaging scenarios [53,99].

### 4.3.3. Limited Availability of Labeled Data

Biomedical image datasets with extensive labeled data for supervised training are often limited. DDPM's performance can be affected when trained on small datasets, potentially leading to overfitting [100].

### 4.3.4. Interpretability and Explainability

DDPM, like many other deep learning models, can be seen as a "black-box" model, making it challenging to interpret and explain its decisions. Understanding the reasoning behind segmentation results is critical in biomedical applications [85,94].

### 4.3.5. Sensitivity to Image Characteristics

DDPM may exhibit sensitivity to certain types of biomedical images, especially those with highly complex anatomical structures or specific imaging artifacts. Ensuring robust performance across diverse image characteristics is essential [101,102].

### 4.3.6. Generalization across Modalities

DDPM's ability to generalize across different imaging modalities may be limited. Variations in data distribution and imaging characteristics between modalities can impact the model's adaptability [79,83].

### 4.3.7. Handling Temporal Information

For dynamic biomedical imaging, such as videos or time-series data, DDPM's applicability might be limited. The model's architecture is primarily designed for static image processing and might not capture temporal dependencies effectively [103,104].

### 4.3.8. Real-Time Constraints

Biomedical applications often require real-time or near-real-time processing, especially in clinical settings. DDPM's computational demands may hinder its feasibility for applications with stringent time constraints [105,106].

### 4.3.9. Model Complexity vs. Dataset Size

The intricate nature of DDPM may lead to overfitting on smaller datasets, where the model might capture noise as if it were genuine signal information [107,108].

### 4.3.10. Integration with Clinical Workflow

The successful implementation of DDPM for biomedical image segmentation requires seamless integration into the clinical workflow. Ensuring usability and compatibility with existing systems is a non-trivial task [109].

In conclusion, while DDPM presents a powerful framework for biomedical image segmentation, its implementation is not without challenges. Overcoming these challenges requires a comprehensive understanding of the specific requirements of biomedical applications, careful parameter tuning, and consideration of ethical and interpretability aspects. Addressing these challenges will contribute to unlocking the full potential of DDPM in advancing the accuracy and reliability of biomedical image segmentation [96].

## 5. Discussion and Conclusions

### 5.1. Implications for the Field of Biomedical Image Segmentation

The DDPM holds significant implications for the field of biomedical image segmentation, introducing novel approaches that address various challenges associated with traditional methods. DDPM excels in denoising biomedical images, contributing to enhanced image quality by effectively reducing noise and artifacts. DDPM introduces a probabilistic framework that allows for the quantification of uncertainty in segmentation predictions. Probabilistic models are valuable in medical applications where uncertainty awareness is essential for clinical decision-making and ensuring the robustness of segmentation results. DDPM can be seamlessly integrated into deep learning architectures, combining the denoising capabilities of DDPM with the representational power of deep neural networks. This integration harnesses the strengths of both approaches, providing a more robust and adaptable solution for complex biomedical image segmentation tasks. In medical applications, where transparency and interpretability are crucial, DDPM offers insights into the reliability of segmentation predictions, fostering trust among healthcare professionals. Improved segmentation accuracy is paramount in medical applications, where precise delineation of structures and anomalies is critical for diagnosis and treatment planning. DDPM, when integrated into deep learning architectures, has the potential to advance segmentation accuracy by addressing noise and uncertainties.

DDPM's *implications for biomedical image* segmentation are multi-faceted, encompassing noise reduction, uncertainty quantification, adaptability to diverse modalities, ethical considerations, and potential integration into clinical workflows. As research in this area progresses, DDPM stands as a promising approach contributing to the advancement of accurate and reliable biomedical image segmentation.

### 5.2. The Potential of DDPM in Advancing Medical Imaging Techniques

In conclusion, the DDPM holds significant promise in advancing medical imaging technology. With its unique combination of denoising capabilities and probabilistic modeling, DDPM addresses key challenges in biomedical image segmentation, contributing to improved image quality, uncertainty quantification, and adaptability across diverse imaging modalities. The integration of DDPM into deep learning architectures further enhances its potential, offering a robust and interpretable solution for accurate segmentation in complex medical scenarios.

DDPM's ability to handle limited annotated data, provide transparency in predictions, and mitigate biases aligns with the ethical considerations essential in healthcare applications. As a catalyst for research in hybrid approaches, DDPM sets the stage for innovative solutions that go beyond traditional segmentation methods, advancing the accuracy and reliability of medical imaging technology.

The probabilistic framework of DDPM not only contributes to noise reduction but also fosters a deeper understanding of segmentation results, promoting trust among healthcare professionals. As medical imaging continues to play a pivotal role in diagnosis and treatment, DDPM stands at the forefront of a new wave of technologies that have the potential to reshape the landscape of image analysis, paving the way for more efficient and reliable clinical decision-making.

In the evolving field of medical imaging, DDPM represents a promising paradigm shift, offering a holistic solution that addresses the complexities of biomedical image segmentation. As research and development in this area progress, the potential of DDPM to enhance the accuracy, interpretability, and ethical considerations in medical imaging technology positions it as a key player in shaping the future of diagnostic and therapeutic approaches in healthcare.

## 6. Future Directions and Challenges

In envisioning the future trajectory of DDPM-based biomedical image segmentation, several avenues present exciting prospects and formidable challenges. One notable avenue

involves the continued integration of DDPM with deep learning architectures, such as convolutional neural networks (CNNs) and generative adversarial networks (GANs), with the overarching goal of enhancing segmentation accuracy for biomedical images. Another promising research direction involves the development of hybrid models that amalgamate DDPM with other probabilistic models or conventional image processing techniques, strategically addressing nuanced challenges in biomedical image segmentation. Given the critical nature of biomedical applications, there is a growing emphasis on advancing DDPM variants to improve interpretability and explainability, ensuring that segmentation results are readily understandable for healthcare professionals. The extension of DDPM-based segmentation approaches to handle three-dimensional (3D) biomedical images and multimodal datasets emerges as a crucial frontier, particularly in medical imaging where 3D representations and diverse imaging modalities are prevalent. Robustness to noisy data and limited annotations poses a significant challenge, prompting research efforts to fortify DDPM-based models against these real-world intricacies. Additionally, exploring transfer learning techniques and pre-training strategies aims to leverage knowledge from other domains, enhancing the generalization capabilities of DDPM-based segmentation models. The incorporation of uncertainty estimation techniques within DDPM models is another avenue, contributing confidence intervals to segmentation results and bolstering the reliability of applications in clinical settings. Optimization for real-time applications in clinical environments, considering computational efficiency and resource constraints, is a critical aspect that researchers are poised to address. Lastly, a noteworthy trend involves an increased focus on clinical validation and adoption of DDPM-based segmentation methods, with more studies demonstrating their efficacy in real-world medical scenarios, thus bridging the gap between research and practical healthcare applications.

**Author Contributions:** Formal analysis, Z.L. and C.M.; investigation, W.S.; resources, W.S. and M.X.; data curation, Z.L. and C.M.; writing—original draft preparation, Z.L.; writing—review and editing, Z.L. All authors have read and agreed to the published version of the manuscript.

**Funding:** This research received no external funding.

**Institutional Review Board Statement:** Not applicable.

**Informed Consent Statement:** Not applicable.

**Data Availability Statement:** Not applicable.

**Conflicts of Interest:** The authors declare no conflicts of interest.

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
