# Peer review of "Biomedical Image Segmentation Using Denoising Diffusion Probabilistic Models: A Comprehensive Review and Analysis"

_applsci, doi:10.3390/app14020632_

Round 1

Reviewer 1 Report

Comments and Suggestions for Authors

This paper presents a comprehensive review on the current state, challenges, and future prospects of utilizing of biomedical image segmentation techniques addressing issues related to the uncertainty and variability in imaging data and sharing the common ground of being based on probabilistic approaches.

I believe that it is remarkable that the authors did not only take into consideration deep-learning based methods to address segmentation tasks, but they focused their attention on other techniques which can indeed be useful for researchers, practitioners, and healthcare professionals; in my opinion, this is a significant strength of the proposed work.

The review is quite complete, but I do believe that there are some missing techniques that should be mentioned, at least in the Introduction, for better picturing the context. I refer to techniques which leverage on the use of graphs and statistical signal processing, such as Markov Random Fields. For authors’ convenience, few reference that are suggested and should be cited follow. Of course, they should/could not limit themselves to these works but can expand to other ones.

[1] Panagiotakis, C., Papadakis, H., Grinias, E., Komodakis, N., Fragopoulou, P., & Tziritas, G. (2013). Interactive image segmentation based on synthetic graph coordinates. Pattern Recognition, 46(11), 2940-2952.

[2] Zhao, Q. H., Li, X. L., Li, Y., & Zhao, X. M. (2017). A fuzzy clustering image segmentation algorithm based on hidden Markov random field models and Voronoi tessellation. Pattern Recognition Letters, 85, 49-55.

[3] Filali, H., & Kalti, K. (2021). Image segmentation using MRF model optimized by a hybrid ACO-ICM algorithm. Soft Computing, 25(15), 10181-10204.

[4] Trombini, M., Solarna, D., Moser, G., & Dellepiane, S. (2023). A goal-driven unsupervised image segmentation method combining graph-based processing and Markov random fields. Pattern Recognition, 134, 109082.

In my opinion, once also these techniques have been mentioned, the picture provided by the authors can be considered completed and the review very exhaustive.

Comments on the Quality of English Language

An English language editing by a native English speaker may be useful.

Reviewer 2 Report

Comments and Suggestions for Authors

Dear Authors,

Thank you for submitting this article. It has a natural flow and interest to the readers. I have few questions which I am attaching here, and I would appreciate it if you could address them.

Reviewer 3 Report

Comments and Suggestions for Authors

This study is an overall survey paper on the DDPM model of biomedical image. This is an excellent study that presented various analyzes and future research directions based on a very rich references.

In order to improve the completeness of this study, please modify the following.

-The abstract is described somewhat abstractly. In particular, please clearly describe the contribution of this study.
-Present the summary of the existing methods introduced in Section 2.1 in a short table format at the end of Section 2.1
-Describe Chapter 5 contents in full sentence format rather than short sentence format.
-It is recommended to exchange the positions of Chapter 5 and Chapter 6. In other words, it would be better for Chapter 5 to be described after Chapter 6.
